

# Integrating hybrid transfer learning with attention-enhanced deep learning models to improve breast cancer diagnosis

Sudha Prathyusha Jakkaladiki and Filip Maly

Faculty of Informatics and Management, University of Hradec Králové, Hradec Kralove, Hradec Kralove, Czech Republic

## ABSTRACT

Cancer, with its high fatality rate, instills fear in countless individuals worldwide. However, effective diagnosis and treatment can often lead to a successful cure. Computer-assisted diagnostics, especially in the context of deep learning, have become prominent methods for primary screening of various diseases, including cancer. Deep learning, an artificial intelligence technique that enables computers to reason like humans, has recently gained significant attention. This study focuses on training a deep neural network to predict breast cancer. With the advancements in medical imaging technologies such as X-ray, magnetic resonance imaging (MRI), and computed tomography (CT) scans, deep learning has become essential in analyzing and managing extensive image datasets. The objective of this research is to propose a deep-learning model for the identification and categorization of breast tumors. The system's performance was evaluated using the breast cancer identification (BreakHis) classification datasets from the Kaggle repository and the Wisconsin Breast Cancer Dataset (WBC) from the UCI repository. The study's findings demonstrated an impressive accuracy rate of 100%, surpassing other state-of-the-art approaches. The suggested model was thoroughly evaluated using F1-score, recall, precision, and accuracy metrics on the WBC dataset. Training, validation, and testing were conducted using pre-processed datasets, leading to remarkable results of 99.8% recall rate, 99.06% F1-score, and 100% accuracy rate on the BreakHis dataset. Similarly, on the WBC dataset, the model achieved a 99% accuracy rate, a 98.7% recall rate, and a 99.03% F1-score. These outcomes highlight the potential of deep learning models in accurately diagnosing breast cancer. Based on our research, it is evident that the proposed system outperforms existing approaches in this field.

## INTRODUCTION

Breast cancer is the most lethal disease in women and remains a severe public health concern in many countries (*The Early Breast Cancer Trialists' Collaborative Group (EBCTCG), 1990*). The American Cancer Society's projections for 2017 anticipated that more than 250,000 women would receive a diagnosis of invasive breast cancer, with over 40,000 succumbing to the disease (*Smith et al., 2017*). Given its complexity and diverse clinical

Corresponding author
Sudha Prathyusha Jakkaladiki,
sudha.jakkaladiki@uhk.cz

outcomes (*Rakha et al., 2010*; *Rivenbark, O'Connor & Coleman, 2013*), different cases of this intricate illness exhibit molecular, behavioral, and physical variations, influencing therapeutic responses.

Invasive breast cancer has become progressively challenging to diagnose and treat, primarily due to its intricate nature and varying clinical manifestations (*Martin et al., 2005*). Thus, having the ability to accurately predict cancer prognosis can significantly benefit both patients and doctors in determining the most suitable treatment approach for individuals with breast cancer. Prognostication is particularly important in clinical practice, especially when dealing with patients with a low chance of survival. Clinicians regularly rely on prognostic prediction information to facilitate clinical decision-making (*Sun et al., 2007*; *Gevaert et al., 2006*), determine a patient's eligibility for specific treatment programs (*Xu et al., 2012*), and plan, conduct, and evaluate clinical trials with the availability of precise prognosis estimates. When a patient is expected to have limited survival time, doctors can offer them the opportunity to consider their treatment options and make realistic preparations for their end-of-life care (*Stone & Lund, 2007*).

People visit an oncologist if they experience any symptoms that might be related to cancer. The oncologist has various tools, such as mammograms, breast magnetic resonance imaging (MRI) scans, breast ultrasounds, breast X-rays, and tissue biopsies, to diagnose and detect breast cancer early. Regular sentinel node biopsies are performed for patients with confirmed breast cancer to check for malignant cells in the lymph nodes. Machine learning algorithms can be employed to classify tumors as either benign or malignant.

An early breast cancer diagnosis can improve survival rates and better prognoses, facilitating timely therapy for patients (*Sun et al., 2017*). Moreover, patients with benign tumors can avoid unnecessary procedures. In the realm of healthcare, data mining techniques have the potential to enhance patient treatment quality, reduce healthcare costs, and even save lives. Deep learning approaches are particularly effective in handling complex and extensive datasets, outperforming traditional machine learning algorithms in accuracy. The research on breast cancer currently concentrates on several datasets related to the subject, furthering our understanding and potential for better diagnosis and treatment.

The rapid growth of breast cancer cases underscores the significance of investing in research and developing innovative approaches for its treatment. This drive urges scientists to discover new methods for swift and accurate diagnoses, ultimately leading to improved patient care and extended lives (*Hamidinekoo et al., 2018*; *Kousalya & Saranya, 2021*). The key to early and consistent detection lies in revising preliminary diagnostic information and gathering relevant data from previous findings, which can be enhanced through medical imaging and deep learning (DL) strategies. Medical imaging plays a critical role in clinically diagnosing diseases, evaluating therapies, and detecting abnormalities in various parts of the body, including the eye (*Akbar et al., 2018*), brain (*Rajinikanth et al., 2017*), breast (*Fonseca et al., 2015*), and stomach (*Houssein et al., 2021*). The primary objective of medical imaging research is to classify affected organs' location, size, and characteristics, offering a promising means of extracting valuable information from extensive datasets. Medical imaging techniques such as mammograms, histopathological

images, MRI scans, ultrasounds, and thermograms are commonly employed in diagnosing breast cancer (*Houssein et al., 2021*).

Over the past decade, thermal imaging (or thermography) has shown tremendous promise in early breast cancer diagnosis. Thermography photos reveal physiological changes that can assist in other forms of diagnostics (*Hossam, Harb & Abd El Kader, 2018*). Another tool used by young women for identifying breast cancer is ultrasound imaging. However, this method may have difficulty detecting microcalcifications and deeper breast tissue due to noise levels (*Qi et al., 2019*), MRI screening, along with ultrasonography and thermography, is yet another method employed for early cancer detection. Magnetic resonance imaging (MRI) enables the creation of exact three-dimensional (3D) transverse images, surpassing X-rays in accuracy (*Houssein et al., 2021*). Mammography stands as a crucial diagnostic imaging technique and serves as the gold standard for breast cancer screening, having demonstrated a significant reduction in breast cancer mortality. As an X-ray examination, mammography is widely regarded as the most reliable and precise tool for detecting breast cancer (*Dhawan, 2011*). In this study, breast cancer was categorized based on mammograms. Our methodology combines DenseNet and a convolutional neural network (CNN), leveraging the strengths of both architectures. This fusion allows us to achieve better classification accuracy and diagnostic performance compared to existing methods on the same dataset. One key innovation in our methodology is the fusion of feature maps from DenseNet and CNN at a standard layer. This approach captures a richer set of features, enhancing the model's ability to differentiate between benign and malignant breast tumors. This feature fusion technique is unique to our work and directly impacts the model's performance.

The notable contributions of the studies, as mentioned above,

1. Examine the Fusion feature fusion methods for better feature extraction.
2. With an accuracy of 99%, the suggested hybrid transfer learning model (a combination of CNN and DenseNet) outperforms state-of-the-art deep learning methods
3. In this work, we employed a pre-trained neural network as a starting point for our model development, leveraging the features and representations learned from diverse datasets during its initial training. Subsequently, we fine-tuned and adapted the model to our specific task of tumor saliency detection using our dataset. This process allowed us to benefit from the knowledge encapsulated in the pre-trained model while tailoring it to our unique application.
4. The optimization method raises the breast cancer detection rate and boosts the efficiency of the suggested system.

## RELATED WORKS

Numerous researchers have explored artificial intelligence, expert systems, and neural networks to enhance the accuracy of breast cancer (BC) screening. Traditionally, hospitals relied on X-rays for breast cancer diagnosis. Still, recent advancements in intelligent modeling have significantly improved the efficiency and accuracy of mammography imaging, leading to the widespread replacement of X-rays. The datasets utilized in *Kiyan &*

*Yildirim (2004)* and *Gonzalez-Angulo, Morales-Vasquez & Hortobagyi (2007)* were obtained from the Kaggle repository. These studies presented a model aimed at determining the likelihood of someone having breast cancer and aiding in early detection and diagnosis. The researchers in these papers compared the effectiveness of various models for predicting breast cancer, including Naive Bayes classifiers and logistic regression. Through machine learning methodologies, their research demonstrated an accuracy ranging from 52.63% to 98.24% in predicting breast cancer illness. Over the past few decades, scientists have dedicated their efforts to studying thermographic breast cancer diagnosis using machine learning techniques. Some researchers have focused on identifying tumor size and location, while others have honed in on different features, such as attainment protocols and breast quadrants. Deep learning, a type of machine learning employing CNNs with multiple hidden layers (*Wehle, 2017*), has shown promising results. With a training dataset, deep learning can automatically extract relevant features. Researchers have made significant progress utilizing CNNs to detect breast cancer in recent years. Interestingly, CNNs were not frequently employed for breast cancer detection using thermal imaging, possibly due to the computational load or the efficiency of CNNs when compared to texture or statistical features (*Zuluaga-Gomez et al., 2021*). Nonetheless, convolutional neural networks have emerged as one of the most promising tools for pattern identification in recent years.

*Tiwari et al. (2020)* and *Naji et al. (2021)* employed a deep learning methodology, a form of deep learning known as convolutional neural networks (CNNs), to classify breast cancer mammography pictures from the public dataset BreakHis. The proposed technique utilized a feedforward network and preliminary trials, where picture patches were extracted for CNN training and then combined for final classification. The accuracy rates for classifying screening mammograms using CNNs were notably high, with the highest accuracy achieved on the Digital Database for Screening Mammography's digitized film mammograms reaching 88% in an independent test set.

In research by *Agnes et al. (2020)* and *Joo et al. (2004)*, the primary objective was to evaluate how well artificial neural networks (ANNs) could categorize tumors into distinct prognostic groups based on gene expression profiles. Small round blue cell tumors (SRBCTs) were used to train the ANN, as clinical practitioners frequently encounter diagnostic challenges related to these four categories of cancer. The ANN correctly classified all samples, and the study identified crucial genes associated with each classification. The experimental findings suggested that the novel approaches introduced in the research might significantly enhance sample classification accuracy and consistency of selection outcomes. As a result, the new algorithms improved the categorization accuracy to almost 99%.

*Chiang et al. (2018)* suggested a computer-assisted screening method for identifying tumors in breast ultrasonography. The technique utilizes a 3D convolutional neural network and a weighted sum of the best candidates. Initially, a sliding window technique is applied to isolate the relevant volumes, and then a 3D convolutional neural network estimates the likelihood of malignancy in each volume of interest. Those with a high likelihood estimate are flagged as potential tumor patients, and their situations may intersect. An original method was developed to accumulate candidates and handle overlaps, ranking

potential candidates based on the likelihood of tumor formation during the aggregation process. Experimental results with 171 tumors showed that the suggested model achieved 95% sensitivity in less than 22 s, demonstrating its superior efficiency compared to current methods. *Agnes et al. (2020)* proposed the multiscale all CNN (MA-CNN) model for breast cancer detection. In this model, mammograms were categorized using a CNN-based method, and multiple scale characteristics extracted from mammography pictures enhanced the CNN classifier's performance. The suggested MA-CNN model proved to be a valuable tool for the early identification of breast cancer from mammograms, as demonstrated by experimental findings, which involved classifying mammograms into benign and malignant categories. Additionally, *Houssein et al. (2021)* developed a deep learning (DL) technique for modeling breast cancer detection, although further details about the DL technique were not provided in the provided text.

Chronic care involves managing and treating long-term medical conditions or diseases (*Shen et al., 2023*). Metal–organic frameworks are porous materials with a highly ordered structure, typically composed of metal ions or clusters connected by organic ligands (*Zeng et al., 2020*; *Liu et al., 2021*). It includes patient vital signs, lab results, ECG readings, or other medical data that can vary over time (*Sun et al., 2023*). This method is likely designed to assist in medical image analysis, diagnosis, or research, where identifying images with similar features can be valuable for tasks like disease detection or treatment planning, particularly in lung-related medical conditions (*Zhuang et al., 2022*; *Zhuang, Jiang & Xu, 2022*). This approach likely involves matching key features or points in the images to align them properly in the mosaic. It provides a more comprehensive view of the area being examined during the endoscopy procedure (*Zhang et al., 2022b*; *Lu et al., 2023b*) This suggests using a deep learning approach involving a neural network for matching and tracking soft tissue features (*Lu et al., 2023a*). This can be a crucial tool in cancer research and medical diagnostics, enabling automated analysis of blood samples to identify and count circulating tumor cells, which can provide valuable information about a patient's cancer status and progression (*He et al., 2020b*; *Xie et al., 2021*; *He et al., 2020a*). This part implies that the overarching goal of the process is to facilitate the repair of the endothelium in the context of hypertension, a condition characterized by high blood pressure and associated vascular damage (*Li et al., 2021*; *Zhang et al., 2022a*). These networks can learn from and predict large, complex datasets. Deep learning has widespread application across various fields, including image recognition, natural language processing, and medical research (*Zhu et al., 2021*; *Gao et al., 2022*). This refers to a long non-coding RNA (lncRNA) called PVT1. LncRNAs are RNA molecules that do not encode proteins but play vital roles in cellular regulation (*Chang et al., 2019*; *Wen et al., 2015*). The primary objective of this research or framework is to determine or predict the specific tissue or organ within the body from which a cancer sample has originated (*He et al., 2020c*; *Wang et al., 2023*; *Jiang et al., 2022*). Haplotypes are genetic variations or alleles that are commonly inherited together. In this case, two specific haplotypes, GCA and ACA, within the ESR1 gene are under examination (*Liu et al., 2022*; *Tang et al., 2022*).

The suggested model for mammography screening employs end-to-end training methods, requiring labels at the picture level throughout the system's resting periods

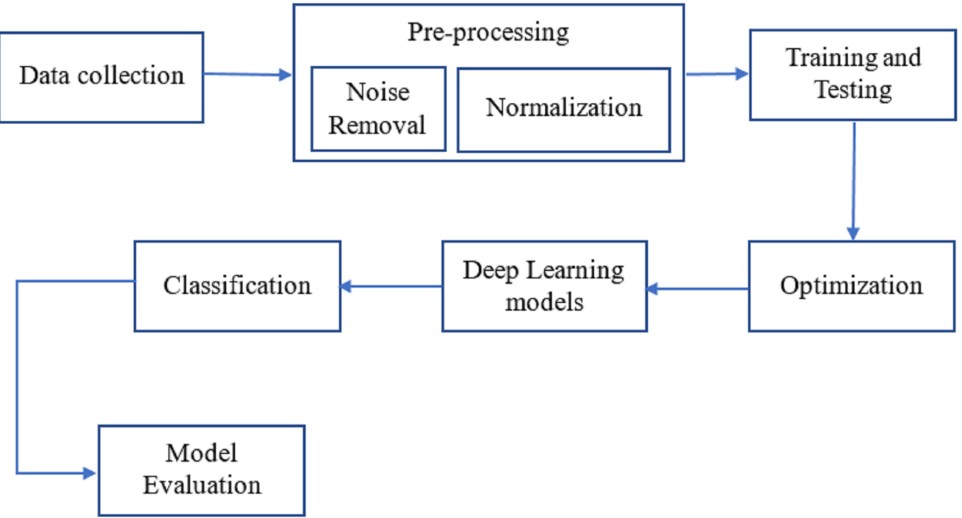

**Figure 1** **The workflow of the proposed methodology for the process of finding breast cancer.**

after the first training phase. This model was trained and tested on a dataset for categorizing mammography images using CNN. The findings indicated that the proposed model outperformed previous efforts in heterogeneous mammography, particularly regarding accuracy. In another study by *Zhou et al. (2018)*, a radionics approach based on convolutional neural networks (CNNs) was introduced for identifying breast cancer. The proposed technique utilized shear-wave elastography data to train a CNN and extract morphology-related features. The model demonstrated high accuracy with training on 540 images, 318 of which were cancerous and 222 benign. Experimental data showed a sensitivity of 96.2%, a specificity of 95.8%, and an overall accuracy of 95.8%. Furthermore, *Qi et al. (2019)* conducted a study highlighting how multiparametric magnetic resonance imaging improved radiologists' ability to diagnose breast cancer. This research used a pre-trained CNN model to extract structures from 927 photos. CNN characteristics were then employed to train an SVM classifier, successfully identifying healthy and cancerous tissue. The study explored the impact of adding more fusion to the mix and discovered that combining features, images, and classifiers played a crucial role in improving accuracy.

## METHODS AND MATERIALS

As can be seen in Fig. 1, breast cancer detection is done. The DenseNet and feature fusion are two important innovations we tested in this method. The effectiveness of neural networks motivates using Dense Net for perfect classification.

### Dataset

The data was obtained from the Kaggle data repository, comprising 7,909 microscopic photos of breast cancer tissue from 82 individuals in the BreakHis dataset. These images were captured at magnifications of $40X$, $100X$, $200X$, and $400X$. Among the samples, there were 2,480 healthy samples and 5,429 cancerous ones. The dataset development involved

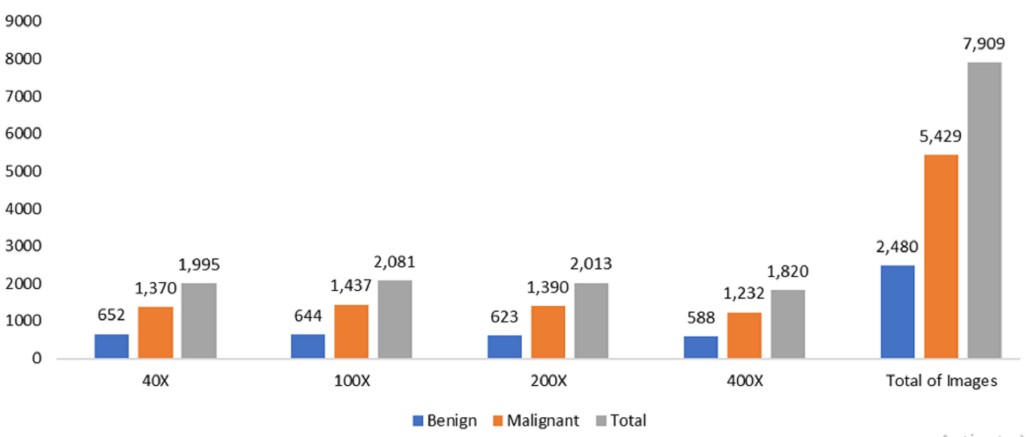

**Figure 2** Data distribution from BreakHis dataset.

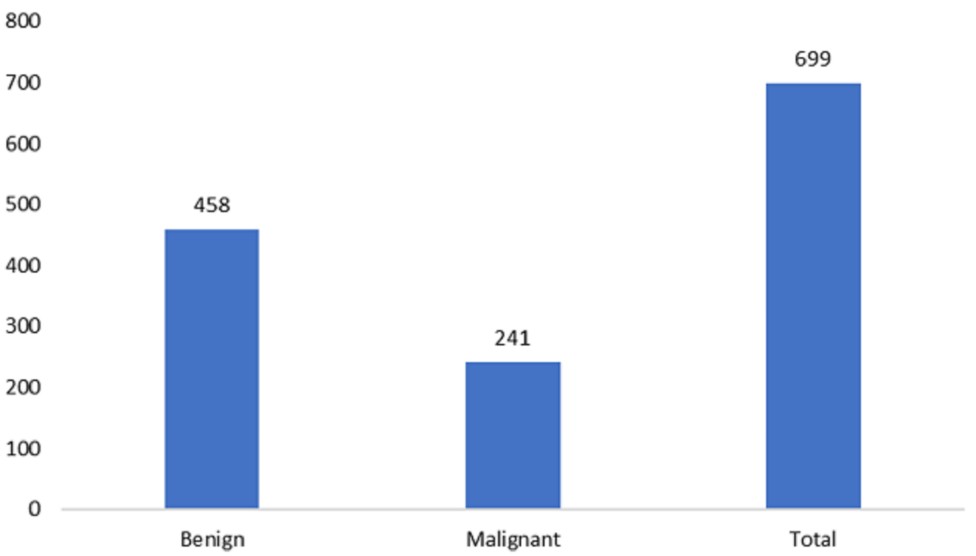

**Figure 3** Data distribution from the Wisconsin-Breast Cancer (Diagnostics) (WBC) dataset.

collaboration with the P&D Laboratory of Pathological Anatomy and Cytopathology in Parana, Brazil.

The measurements for breast cancer patients are recorded in the original Wisconsin Breast Cancer (WBC) dataset, available directly from the UCIML library. The dataset consists of both noncancerous and cancerous varieties. Specifically, this dataset's malignant category, containing outliers, comprises 21 points. In contrast, the inliers belong to the safe group. The Wisconsin cancer dataset includes 699 cases, with 458 noncancerous (accounting for 65.5% of the cases). Figures 2 and 3 illustrate the data distribution from the datasets.

## Pre-processing

As discussed previously, noise reduction from input pictures is a crucial issue in medical imaging. Maintaining picture edges' maximum sharpness and integrity is vital during the noise-reduction process. Each pixel in the output is determined by calculating the median of the brightness values of the surrounding pixels in the input (*Zhang & Hong, 2019*). Median filtering establishes a pixel's significance by averaging its neighboring pixels' importance. The central filter is not highly influenced by outliers, enabling it to eliminate these values without compromising the overall image quality. This filter preserves edge form and position while reducing light intensity variance (*Sharifrazi et al., 2021*). The m x n neighborhood filter utilizes an ascending sort on the sorted data before replacing the center pixels with a new set of values. Additionally, the median filter effectively filters out salt and pepper noises (*Song, Jia & Ma, 2019*). For this reason, we applied this filter as the first step in the processing pipeline to the input photos. The median value of the surrounding pixels is used to replace the original one in median filtering.

$$a_{(x,y)} = median(b_{m,n} : (m,n)) \in \tau \tag{1}$$

where $(x,y)$ is the set of closest neighbors defined by $\tau$. The filter size in this example is $5 \times 5$ pixels.

After eliminating the background noise, the images must be standardized by being scaled between 0 and 1 to make the dataset more stable. Here, we employed the min-max technique of normalization. Given the following restriction,

$$M_n = \left[A \subseteq \mathbb{R}^n\right] \to \left[x, \ldots\ldots y\right] \tag{2}$$

on the grayscale picture and its dimensions, the normalized image, $M^*$, can be described as follows.,

$$M^* = x_{new} + \frac{y_{new} - x_{new}}{y - x} \times (M - x) \tag{3}$$

in where x and y are image intensities.

$$M^* = \left[A \subseteq \mathbb{R}^n\right] \to \left[x_{new}, \ldots\ldots y_{new}\right] \tag{4}$$

in where $x_{new}$ and $y_{new}$ are normalized image intensities.

Various pre-processing techniques were applied before inputting the pictures into the customized variational deep-learning algorithm. All micro pictures in BreakHis are saved as PNGs with an 8-bit resolution per channel and three RGB channels. Using machine learning methods, the size of numerous high-resolution photos was reduced to 224 by 224. The proposed technique converts the collected images into Numpy arrays, facilitating faster model training with minimal space requirements. Several data augmentation approaches were utilized to reinforce the model to address overfitting issues. The dataset size increased from 7,909 to 54,403 through the data preparation method. In the pre-processing phase, a zoom factor of 2, a rotation angle of 90 degrees, a shearing factor of 0.5, and a shifting factor of 0.4 were applied to both the width and height of the image.

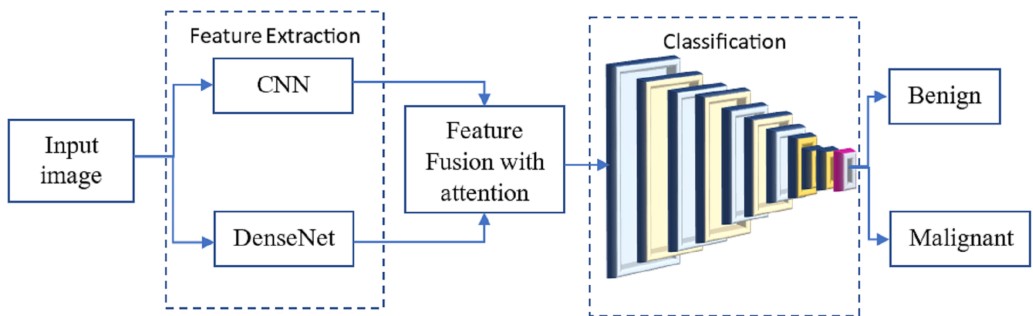

**Figure 4** **The proposed methodology of the breast cancer prediction with feature fusion on CNN and DenseNet.**

## Proposed methodology

Many researchers have investigated the use of CNN for breast cancer diagnosis, an essential topic in medical picture processing. One kind of CNN that has performed well in picture classification is called DenseNet. The term "feature fusion" describes merging features from several models to boost performance. We may experiment with a DenseNet and a CNN to identify breast cancer in this scenario. One approach to feature fusion involves merging the feature maps of these two models at a standard layer, and the combined output is then passed through a fully connected layer to produce the final classification result. Alternatively, selective feature fusion can be achieved using attention techniques. The proposed model, as depicted in Fig. 4, incorporates these feature fusion strategies to enhance breast cancer diagnosis performance.

The resulting model can be trained on numerous annotated mammograms through supervised learning. Both DenseNet and CNN weights are adjusted concurrently to achieve optimal performance during training. Once the model is introduced, it can be utilized to determine whether a new mammogram is cancerous or benign.

### Convolutional neural network

CNNs uncover hidden patterns in images by convolving over the picture. In the early stages of CNNs, the network identifies simple patterns like straight lines and sharp angles. However, as we delve deeper using our neural net, we can capture more complex features. This characteristic makes CNNs highly effective in detecting objects in images. The suggested technique employs convolutional neural networks to identify breast cancer based on photographic evidence.

Figure 5 illustrates the primary layers constituting a CNN's architecture: the convolutional, pooling, and fully connected layers. In the first layer, neurons connected to nearby areas compute their output. Each output is determined by the dot product of the weights and the corresponding area. Standard filter sizes for input images are typically $3 \times 3$, $5 \times 5$, or $8 \times 8$ squares. By moving a window over the image, these filters learn patterns that appear throughout the picture. The stride represents the distance between successive filters. If the stride hyperparameter exceeds the filter dimension, the convolution expands

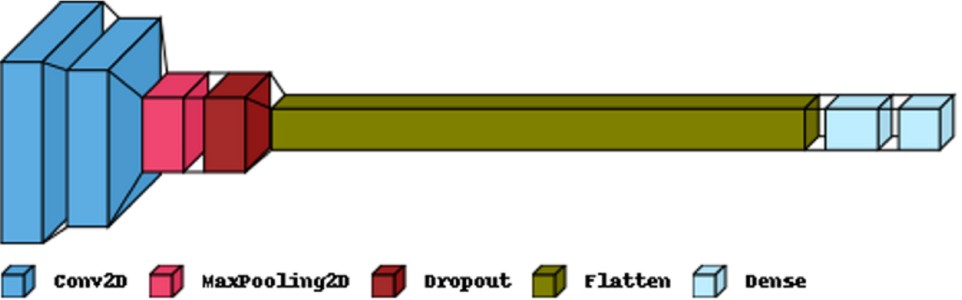

**Figure 5** The proposed CNN model structure for the feature extraction.

**Table 1** The parameter setting for the convolutional neural network model.

| Layer (type) | Output shape | Param |
| --- | --- | --- |
| Conv2D | (22, 22, 32) | 1,568 |
| Conv2D | (19, 19, 32) | 16,416 |
| MaxPooling2 | (9, 9, 32) | 0 |
| Dropout | (9, 9, 32) | 0 |
| Flatten | (2,592) | 0 |
| Dense | (256) | 663,808 |
| Dense | (2) | 514 |
| Total params: 682,306 | | |
| Trainable params: 682,306 | | |
| Non-trainable params: 0 | | |

to overlap adjacent windows. Table 1 comprehensively describes neural networks (NNs) in terms of their parameter numbers.

The CNN model is commonly employed for data training in medical image analysis, examination, and various applications. CNN is crucial in the successful computer aided diagnosis (CAD) systems used for medical imaging. This makes CNN a suitable choice for our intended application of breast cancer detection, although there is still potential for improvement. In a typical CNN, the kernels consist of hundreds of neurons arranged in multiple layers. Small kernels are used to maintain a constant depth for the input picture. Neurons are connected to a receptive field, representing a relatively narrow region. Especially when dealing with high-dimensional input images, connecting all neurons directly to the previous outputs becomes exceedingly challenging. As a result, alternative approaches are utilized to address this issue.

The design of our model is built upon the Sequential model, allowing the stacking of input and output layers sequentially. The convolution layer employs convolutional filters to perform a full-image scan of the input data. Its hyper-parameters include the filter size and stride (the distance between two successive receptive filters). The result of this operation is referred to as an activation map or feature map. We first create a 2D convolutional layer to process the input breast pictures. The number of output channels (16 channels in our

case) is the first input to the convolution layer function. We utilize a spatial convolution algorithm with a 3x3 filter kernel and a stride of 1, allowing the kernel to slide horizontally and vertically across the entire image. Throughout our trials, we experimented with various kernel sizes, ranging from 1 to 7, to assess their impact on the results. We observed that the 1x1 kernel size limited the system's performance. We have considered padding to ensure that the filter and stride defined by this model completely cover the input picture.

The input, denoted by x, has been used to activate a system according to the rectified linear activation function (RELU), which is written as,

$$f(x) = (0, x) \tag{5}$$

The spatial size of the feature map is shrunk *via* max pooling. The Maxpooling function downsamples the input depiction by selecting the most significant value along each dimension of the feature's axis within the window given by pool_size. Two further convolutional layers are added, each with 32 or 64 output channels, and the process is repeated. Initially, we employed a $2 \times 2$ maximum pooling filter. Our approach to fixing the degradation issue is based on the same idea as the deep residual learning architecture utilized. It operates on recursive units of $1 \times 1$, $3 \times 3$, and $1 \times 1$ convolution filters. The activation of each feature map was calculated using a global average pooling method.

### DenseNet model

To test and analyze our dataset, we employed the DenseNet CNN framework. DenseNet offers several advantages over previous pretraining CNN techniques, including improved handling of the vanishing-gradient problem, fewer parameters, increased feature reuse, and enhanced feature propagation. CNN is a series of feedforward layers that utilize convolutional filters and pooling layers. The CNN employs multiple fully connected layers after the final pooling layer to convert the 2D feature maps learned in earlier layers into a 1D vector for classification. The notation for this is,

$$F(y) = f_m\big(f_m - 1\big(\dots\big(f_1(y)\big)\big)\big) \tag{6}$$

Our method employs a total of $m$ hidden layers. $y$ is the input data and $f_m$ is the function of layer m. The convolutional layer in a standard CNN model is a function $f$ that takes in a series of convolutional kernels $(l_1, l_2 \dots l_k)$, each performing a different task. For each $k$, $l_k$ represents a linear function in the kth kernel:

$$l_k(M, N) = \sum_{q=-x}^{x} \sum_{r=-y}^{y} \sum_{s=-h}^{p} W_k(q, r, s)\, Y(y - q, z - r, x - s) \tag{7}$$

The input y's input pixel is defined by $(q, r, s)$. The kernel weight is denoted by $W_k$.

### Attention mechanism

In the process of making predictions, a deep learning model can pinpoint relevant aspects of the input through the use of attention mechanisms. The attention mechanism functions by assigning varying weights to different input features based on their predictive importance. Subsequently, the prediction is made using the weighted sum of these input characteristics.

This approach enhances the model's performance by allowing it to focus on the most crucial elements while disregarding less significant ones.

The attention mechanism is a vital concept in deep learning, especially when dealing with long input sequences and the risk of information loss due to a fixed intermediate vector length. Since its development for the seq2seq paradigm in Natural Language Processing (NLP), the attention mechanism has gained rapid adoption in various domains. The output of the attentional mechanism can be expressed as a way to effectively address this issue as,

$$Attention(X, Y, Z) = softmax\left(\frac{XY^T}{\sqrt{k_m}}\right) \tag{8}$$

where, $K = IW_k, \quad K \in (X, Y, Z)$ and $I$ is the input, $W_k$ is the learnable weight matrix. The dimension of the keys is denoted by $k_m$. The softmax is the activation function, which is calculated as,

$$softmax(y_i) = \frac{e^{y_i}}{\sum_i e^{y_i}} \tag{9}$$

## Classification

A typical CNN model comprises five layers: the convolution layer, the rectified linear activation function (RELU) layer, the max pooling layer, the fully connected layer, and the dropout layer. The convolution layer holds utmost significance in a CNN, utilizing trainable filters whose settings are adjusted in each cycle. The RELU layer is a popular choice in CNN designs as it reduces training time. The Max pooling layer is often employed to manage overfitting and decrease parameter size. In the fully connected layer, neurons form a typical neural network structure. Lastly, the dropout layer is utilized to avoid overfitting.

It is proposed to employ a deep learning model based on a CNN to distinguish between healthy and diseased breast tissue. Figure 6 depicts the network's nine layers, of which the first six are convolutional, and the last three are fully linked. The proposed model has a first layer that filters a 228 × 228 input picture using 64 7 × 7 kernels spaced by a stride of 6 pixels. The depth = 3 of the first layer's kernels determines the number of color channels in the source thermogram. The output of the first layer is sent into the second layer, which filters it using 128 kernels of size 3 × 3 × 64, and then max-pooling is used to increase resilience and decrease computation. The third, fourth, and fifth levels are joined without pooling layers. The size of each kernel in the third layer is 3 × 3 × 128, and there are 256 of them. There are 256, 3 × 3 × 256 kernels in the fourth layer and the same number in the fifth layer. The sixth layer contains 256, 3 × 3 × 256 kernels and is coupled to the fifth layer through a max-pooling layer. Two layers, each with 1024 neurons, are fully linked on top of the convolutional layers. Class size is proportional to the number of neurons in a neural network's last, fully-connected layer.

The "Optimizer RMSprop" was employed to evaluate various activation functions in TensorFlow, the 1D tensors generated by the convolutional layers, and the output "Flatten" layer, all contributing to weight optimization. The remaining dataset images not used in training were utilized during the test phase. The desired image feature vector is obtained

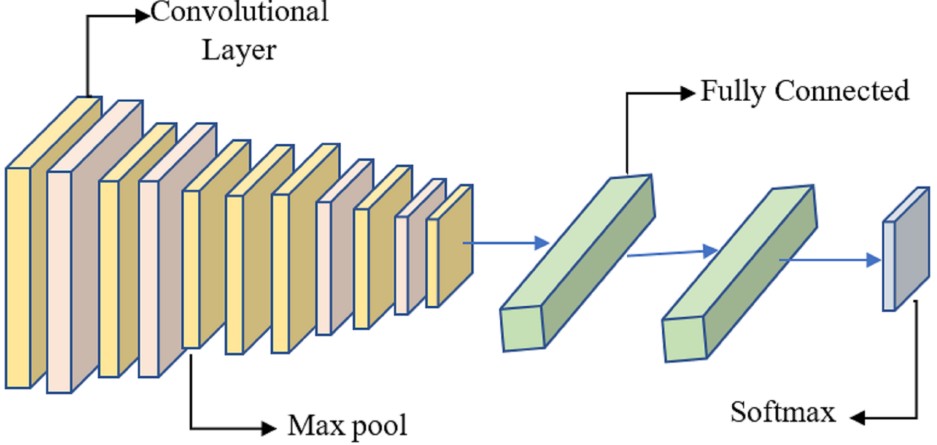

**Figure 6** **System for benign and malignant data classification using fused features from a transfer learning framework.**

from the layer output in this phase. Subsequently, the image feature vector is compared to the feature matrix, and certain deep neural network layers capture specific data sections. However, a set of possibilities is needed for the final judgment of data categorization. To achieve this, the softmax function is often used to normalize the probability values to the interval [0, 1]. Convolutional layer output is determined by,

$$Z_c = \frac{I_s - K_s + 2d}{Stride} + 1 \tag{10}$$

The output of the pooling layer is calculated as,

$$Z_p = \frac{I_s - P_s}{Stride} + 1 \tag{11}$$

where the $I_s$ signifies the size of the input and $K_s$ symbolizes the kernel size, the padding is designated by $d$ and the size of the pooling layer is signified by $P_s$. As noted above, the "Optimizer RMSprop" technique was used to determine the weights optimally. Reducing the cross-entropy is the basis of the system's optimization functioning, which has the following mathematical form.

$$L = \sum_{l=1}^{K} \sum_{p=1}^{J} -v_p^l \, log \, y_p^l \tag{12}$$

The sample number is signified by $K$. The output vector is denoted by $v_p = (0, ....., 0, 1, ...... 1, 0, ......, 0)$. The predicted output is characterized by $y_p$ and the class is formulated by,

$$k_p^l = \frac{e^{g_p}}{\sum_{l=1}^{J} e^{g_p}} \tag{13}$$

With the weight penalty,

$$L = \sum_{l=1}^{K} \sum_{p=1}^{J} -v_p^l \, log \, y_p^l + \frac{1}{2}\mu \sum_{C} \sum_{L} U_{c,l}^2 \tag{14}$$

The total number of layers $L$, and the connections of layer L are denoted by C. The weight is described $U_c$.

### Hyper parameter tuning

This section discusses the hyperparameters associated with the fine-tuned DenseNet model, such as learning rate and batch processing loss. Choosing appropriate parameters and avoiding overfitting and underfitting the model is crucial for achieving the best training and testing results. This paper examines the amount of data lost during training and the accuracy of the test results. The model's divergence begins when it has already surpassed the optimal learning rate range. Ideally, the loss should continue to decrease when the learning rate is selected. Regarding the optimizer's L2 penalty (weight decay), the author suggests using the highest possible learning rate, allowing for faster training than grid-search when using weight decays of 0.01 (the minimum), 0.0001 (the next lowest), and 0.000001 (the highest). To address the overfitting issue, two dropout layers are utilized. During model training, a significant amount of data is lost after the first dropout layer, and progressively less data is lost with each succeeding dropout layer.

## EXPERIMENTAL SETUP

The tests in this study were conducted using Python 3 and a GPU. Keras and the scikit-learn package were utilized to successfully create the optimal deep-learning models. The dataset was divided into two sections: a training dataset consisting of 80% of the total data used for model enhancement and registering cross-validation (CV) results, and a testing dataset consisting of 20% of the total data used for model assessment and registering testing results. We employed two feature-selection techniques based on correlation to narrow down the potential features to eight. Then, the features selected *via* correlation, the features selected *via* univariate, and the features chosen *via* RFE were used as input for the optimal deep learning models. A few settings in the deep learning optimization were adjusted for each set of 32 batches and 100 epochs used in the experiment. The tests were repeated a total of four times.

## RESULT AND DISCUSSION

The recital of the proposed framework is assessed using several numerical indicators, including precision, recall, false-positive rate, true-negative rate ($TR_{NG}$), F1-score, and Matthew's correlation coefficient (MCC). The final result of the confusion matrix determines values for parameters like true positive ($TR_{PS}$), true negative ($TR_{NG}$), false positive ($FP_{PS}$), and false negative ($FP_{NG}$). "$TR_{PS}$" denotes a result where the proposed. All available metrics for assessment are listed below.

$$Accuracy\,(AC) = \frac{TR_{PS} + TR_{NG}}{TR_{PS} + TR_{NG} + FP_{PS} + FP_{NG}} \tag{15}$$

**Table 2  Sixfold cross-validation results (accuracy) on BreakHis and WBC datasets.**

| Dataset/Fold | BreakHis | WBC |
|---|---|---|
| Fold 1 | 99.7 | 99.4 |
| Fold 2 | 99.6 | 99.5 |
| Fold 3 | 99.8 | 99.3 |
| Fold 4 | 99.7 | 99.6 |
| Fold 5 | 99.9 | 99.4 |
| Fold 6 | 99.8 | 99.7 |

$$Sensitivity\,(SE) = \frac{TR_{PS}}{TR_{PS} + FP_{NG}} \tag{16}$$

$$Specificity\,(SP) = \frac{TR_{NG}}{TR_{NG} + FP_{PS}} \tag{17}$$

$$Precision\,(PR) = \frac{TR_{PS}}{TR_{PS} + FP_{PS}} \tag{18}$$

$$Recall\,(RE) = \frac{TR_{PS}}{TR_{PS} + FP_{NG}} \tag{19}$$

$$F1 - score\,(F1s) = 2 \times \frac{PR \times RE}{PR + RE} \tag{20}$$

$$MCC = \frac{TR_{PS} \times TR_{NG} - FP_{PS} \times FP_{NG}}{\sqrt{(TR_{PS} + FP_{PS})(TR_{PS} + FP_{NG})(TR_{NG} + FP_{PS})(TR_{NG} + FP_{NG})}} \tag{21}$$

Table 2 presents the results of a sixfold cross-validation experiment conducted on two datasets: the BreakHis dataset and the WBC dataset. Cross-validation is a widely used technique in machine learning and data analysis to assess the performance and generalization ability of a model on unseen data. Each cross-validation fold involves splitting the dataset into multiple subsets, training the model on some subsets, and testing it on others. This process is repeated multiple times, and the results are averaged to obtain a more robust evaluation of the model's performance.

Table 3 demonstrates that the conceptual approach is superior to the other five models in identifying breast cancer subtypes. Pre-trained DenseNet and ResNet50 models from a five-transfer deep learning framework obtain an estimation accuracy of greater than 95.

The model's training performance is excellent even before it has been fine-tuned. Defrosting the lowest layers pre-trained with other information and re-training the model with our cancer data allows us to fine-tune the model by changing the weights of these

**Table 3   Performance analysis of the baseline models and the proposed model executed on the BreakHis and WBC datasets.**

| Dataset | Model | Precision | Recall | Accuracy | F1-Score | MCC |
|---------|-------|-----------|--------|----------|----------|-----|
| BreakHis | DenseNet | 0.98 | 0.98 | 0.97 | 0.98 | 0.97 |
| | VGG16 | 0.95 | 0.94 | 0.95 | 0.95 | 0.95 |
| | MobileNet | 0.94 | 0.93 | 0.94 | 0.94 | 0.94 |
| | ResNet-50 | 0.96 | 0.97 | 0.96 | 0.96 | 0.97 |
| | CNN | 0.96 | 0.96 | 0.96 | 0.96 | 0.97 |
| | Proposed model | 0.99 | 0.99 | 0.99 | 0.99 | 0.99 |
| WBC | DenseNet | 0.97 | 0.97 | 0.97 | 0.98 | 0.98 |
| | VGG16 | 0.96 | 0.97 | 0.96 | 0.96 | 0.96 |
| | MobileNet | 0.94 | 0.93 | 0.94 | 0.93 | 0.94 |
| | ResNet-50 | 0.96 | 0.97 | 0.96 | 0.96 | 0.95 |
| | CNN | 0.95 | 0.96 | 0.95 | 0.94 | 0.94 |
| | Proposed model | 0.99 | 0.99 | 0.99 | 0.99 | 0.99 |

layers. Once the model has been unfrozen, it is trained at a significantly slower learning rate.

Figures 7A and 7B shows that the training and validation losses remain consistent across iterations. This indicates that with a low learning rate, the model starts to overfit. Further training would result in overfitting, where the model learns only from the characteristics present in the training set, leading to improved validation results. However, despite the model being trained to a certain extent, it performs poorly on real-world data. The previous figures clearly show that, with some adjustments, the model achieves impressive results on the data. The training loss and accuracy measures assess the model's effectiveness. As observed, the model's loss starts near zero and gradually increases to approximately 20 training batches. After that, the loss rapidly drops to almost zero, concluding the training for that cycle. The model maintains a constant momentum of 0.9 throughout the training process. However, after 20 epochs, the learning rate experiences a sharp increase. This learning process reaches a threshold represented by a variable. As the training progresses, the pace slows down again. The increased rate has a regularising effect, pushing the model away from unstable minor local minima and towards more stable wide minima. As we approach the midpoint of the cycle, the learning rate is slowed down in the hope of reaching a steady state. The next step is to search for the lowest values in the region.

The proposed architecture required 50 epochs to train and attained 100% accuracy after ten epochs on the BreakHis dataset and 99% accuracy after 12 epochs on the WBC dataset. Consequently, the overall accuracy of the validation process is 99%. The loss value remains close to 0 with no significant variations during training and validation on the BreakHis dataset, and it is 0.3 on the WBC dataset, as illustrated by the error function graph in Figs. 7C and 7D.

True positives ($TR_{PS}$), true negatives ($TR_{NG}$), false positives ($FA_{PS}$), and false negatives ($FA_{NG}$) are included in a table called the confusion matrix. When both the predicted and observed values for an outcome are positive, the former is referred to as a $TR_{PS}$. When a

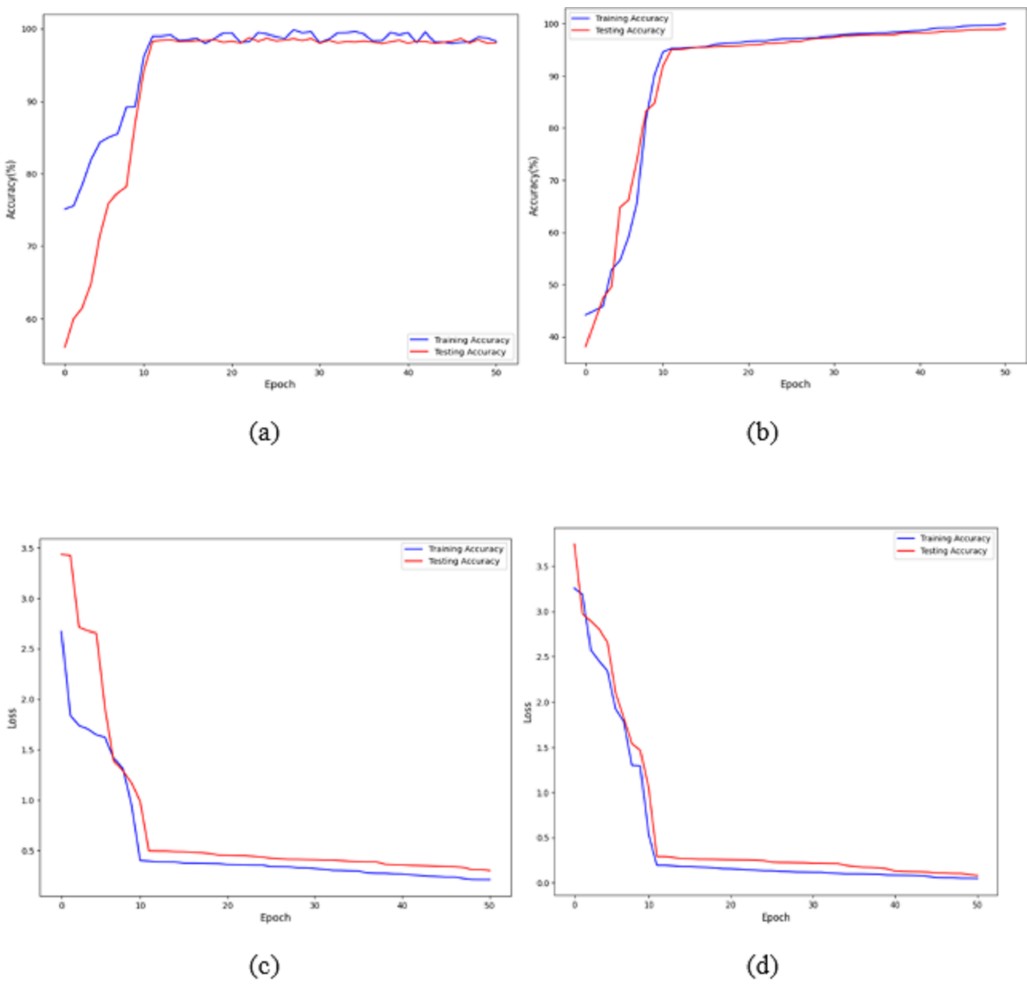

**Figure 7 Accuracy and loss of the proposed model to detect breast cancer: (A, C) BreakHis, and (B, D) WBC.**

model correctly identifies an instance as negative (or 0), and the actual output is negative, we refer to this as a TruN output. When a model predicts a positive (or 1) outcome while the actual outcome is negative, we refer to the expected outcome as $FA_{PS}$. When a model predicts a negative (or 0) outcome while the actual result is positive, the expected result is referred to as $FA_{NG}$. It follows that the model's accuracy improves when the number of $TR_{PS}$ and $TR_{NG}$ increases (or decreases, in the case of $FA_{PS}$ and $FA_{NG}$). True positive indicates a tumour was present on the slide being examined, whereas true negative indicates no cancer. When a tumour is incorrectly diagnosed as not malignant, the result is a false negative. After training on their own, the DenseNet-169 and the improved DenseNet-169 were compared using confusion matrices. The DenseNet 169 confusion matrices are shown in Fig. 8.

Table 4 presents an analysis of the false negative rate (FNR) and false omission rate (FOR) (*Gonzales-Martinez & van Dongen, 2023*) for a set of baseline models and a proposed

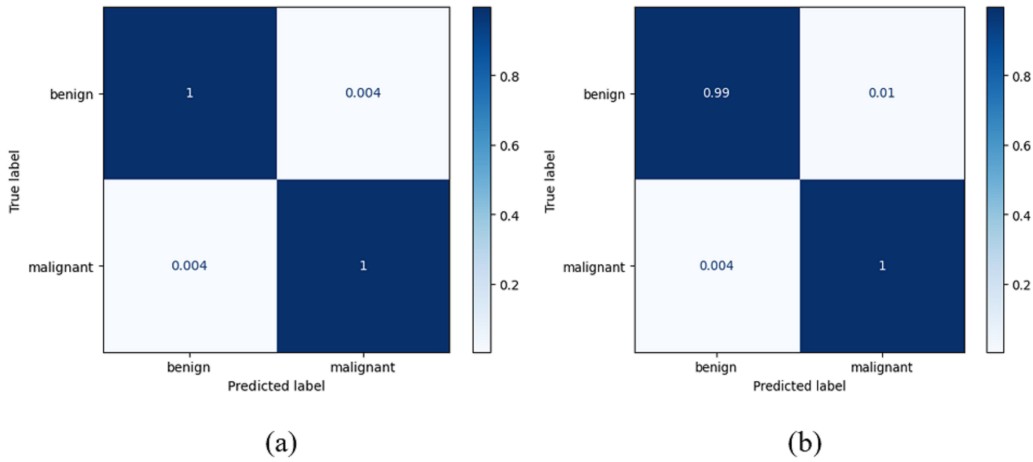

**Figure 8** Confusion matrix for the classification of benign and malignant data from the dataset: (A) BreakHis (B) WBC.

**Table 4** FNR and FOR analysis of the baseline models and the proposed model executed on the BreakHis and WBC datasets.

| Dataset | Model | FNR | FOR |
|---------|-------|-----|-----|
| BreakHis | DenseNet | 0.02 | 0.03 |
| BreakHis | VGG16 | 0.06 | 0.05 |
| BreakHis | MobileNet | 0.07 | 0.06 |
| BreakHis | ResNet-50 | 0.03 | 0.04 |
| BreakHis | CNN | 0.04 | 0.05 |
| BreakHis | Proposed model | 0.01 | 0.01 |
| WBC | DenseNet | 0.03 | 0.03 |
| WBC | VGG16 | 0.03 | 0.04 |
| WBC | MobileNet | 0.07 | 0.06 |
| WBC | ResNet-50 | 0.03 | 0.04 |
| BreakHis | CNN | 0.04 | 0.05 |
| BreakHis | Proposed model | 0.01 | 0.01 |

model when executed on two different datasets: BreakHis and WBC. This analysis provides valuable insights into the models' performance, particularly their ability to correctly identify positive cases and minimise the risk of false negatives and omissions.

In a study conducted by *Martinez & van Dongen (2023)*, a comparison of deep learning algorithms was performed to evaluate error rates. The researchers reported a false negative rate (FNR) of 0.078 and a false omission rate (FOR) of 0.0983. This study introduces Advance Scheduling, a groundbreaking model that outperforms previous findings by achieving significantly lower false negative rate (FNR) and false omission rate (FOR). Specifically, our model demonstrates exceptional performance, with FNR and FOR recorded at an impressive 0.01 each. The findings indicate a notable enhancement in

the precision and dependability of the suggested model compared to the previously evaluated deep learning algorithms.

## CONCLUSION

Most cases can be cured if diagnosed early and treated appropriately. Computer-assisted diagnosis, especially in artificial intelligence, has dramatically facilitated early cancer identification and diagnosis. This research focused primarily on breast cancer, utilising a deep neural network to provide reliable diagnoses and classifications of breast malignancies. The study's contributions include investigating feature fusion techniques for improved feature extraction and creating a hybrid transfer learning model that outperforms current deep learning techniques by 99% accuracy. An attention-enhanced deep learning model was developed to further enhance the system's efficacy and increase the breast cancer detection rate, incorporating historical data on tumour saliency. The results showed that the proposed model achieved outstanding performance, surpassing state-of-the-art methods with 100% accuracy, 99.8% recall, and 99.06% F1-score on BreakHis, and 99% accuracy, 98.7% recall, and 99.03% F1-score on WBC. The evaluated results on FNR and FOR for Breakhis and WBC datasets show that the proposed model achieves fewer false negatives of 0.01 for both datasets. Overall, this study demonstrates the potential of deep learning to enhance the precision and timeliness of breast cancer detection and diagnosis. With further refinement, this technique could significantly save lives by enabling earlier breast cancer diagnosis and treatment. The outcomes of this system show improvement compared to earlier models. However, the system's training time is a bottleneck due to the in-depth training of the neural network. The process will run faster on computers with GPUs than on traditional hardware. Therefore, users are expected to have access to more computationally capable devices for testing and analysing their data. These strategies will play a crucial role in cancer diagnostic and prediction tasks. However, further testing and validation on larger datasets are necessary for clinical use. Their performance might benefit from more investigation into data augmentation techniques, learning in domains like the frequency domain, and implementing innovative designs like graph convolutional networks.

### Funding
This study was financially supported by the Super Doctorand program and SPEV project, run by the faculty of informatics and management, at the University of Hradec Kralove, Czech Republic. The funders had no role in study design, data collection and analysis, decision to publish, or preparation of the manuscript.

### Grant Disclosures
The following grant information was disclosed by the authors:
Super Doctorand program and SPEV project, run by the faculty of informatics and management, at the University of Hradec Kralove, Czech Republic.

## Competing Interests

The authors declare there are no competing interests.

## Author Contributions

- Sudha Prathyusha Jakkaladiki conceived and designed the experiments, performed the experiments, analyzed the data, performed the computation work, prepared figures and/or tables, and approved the final draft.
- Filip Maly conceived and designed the experiments, performed the experiments, analyzed the data, performed the computation work, authored or reviewed drafts of the article, and approved the final draft.

## Data Availability

The data are available at Kaggle at UC Irvine: https://www.kaggle.com/datasets/ambarish/breakhis.

The Diagnostic Wisconsin Breast Cancer Database (William Wolberg, Olvi Mangasarian, Nick Street, W. Street) is available at Wolberg,William, Mangasarian,Olvi, Street,Nick, and Street,W.. (1995). Breast Cancer Wisconsin (Diagnostic). UCI Machine Learning Repository. https://doi.org/10.24432/C5DW2B.

## Supplemental Information

Supplemental information for this article can be found online at http://dx.doi.org/10.7717/peerj-cs.1850#supplemental-information.

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
