# Peer review of "Integrating hybrid transfer learning with attention-enhanced deep learning models to improve breast cancer diagnosis"

_PeerJ Computer Science, doi:10.7717/peerj-cs.1850_

## Round 0.1 · original submission · Major Revisions

Based on the referee reports, I recommend a major revision of the manuscript. The author should improve the manuscript, taking carefully into account the comments of the reviewers in the reports, and resubmit the paper.

Reviewer 2 has requested that you cite specific references. You may add them if you believe they are especially relevant. However, I do not expect you to include these citations, and if you do not include them, this will not influence my decision.

Reviewer 1 ·

Basic reporting

All suggestions are added to the additional comments section.

Experimental design

All suggestions are added to the additional comments section.

Validity of the findings

All suggestions are added to the additional comments section.

Additional comments

The authors presented the article entitled "Integrating Hybrid Transfer Learning with Attention-Enhanced Deep Learning Models to Improve Breast Cancer Diagnosis". The topic is interesting from the healthcare point of view, but the contribution is questionable. Some comments to the authors are as follows.
• The work done in this paper is basic. The same dataset has been used, and other methods were implemented. So why do we need your methodology?
• The dataset has not been shown in this manuscript. Please consider showing a few of the images for better understanding.
• "We developed an attention-enhanced deep learning model to include past information on tumour saliency". When a pre-trained network is used, which is trained and optimized with other datasets, how can you say past information is used for model training?

·

Basic reporting

EDITED

The authors proposed a deep learning (DL) model based on convolutional neural networks (CNN) with an attention mechanism to improve breast cancer diagnosis with medical imaging. The system’s performance was evaluated using the breast cancer identification classification datasets from the Kaggle repository and the Wisconsin Breast Cancer Dataset (WBC) from the UCI repository. The proposed model has superior performance metrics, compared to other architectures, in terms of precision, recall, accuracy, the F1-score, and Matthew’s correlation coefficient (MCC).

* The article is interesting and is written with technically correct text. Clear and unambiguous, professional English is used in general, but some further improvements can be implemented to avoid redundancy, as in line 142, where it says that details about DL techniques "were not provided in the provided text", can, for example, be reduced to "DL techniques were not provided".
* The article can benefit from an additional round of writing polishing, as some typos still exist in the document, for example "CN)" in line 267.
* The article has a professional article structure and the data and Python scripts were shared and accounted for in the article.
* The use of references should be improved by properly quoting the articles, as for example in line 52 it should be "(Sun et al., 2017)", instead of "Sun et al. (2017)".
* I suggest the authors add to their references the article of Gonzales-Martinez and van Dongen (2023), due to the reasons described below in Section 2 (experimental design).

Gonzales-Martinez, R., and van Dongen, D. M. (2023). Deep Learning Algorithms for Early Detection of Breast Cancer: A Comparative Study with Traditional Machine Learning. Informatics in Medicine Unlocked
Volume 41, 2023, 101317

https://www.sciencedirect.com/science/article/pii/S2352914823001636


ORIGINAL
The authors proposed a deep learning (DL) model based on convolutional neural networks (CNN) with an attention mechanism to improve breast cancer diagnosis with medical imaging. The system’s performance was evaluated using the breast cancer identification classification datasets from the Kaggle repository and the Wisconsin Breast Cancer Dataset (WBC) from the UCI repository. The proposed model has superior performance metrics, compared to other architectures, in terms of precision, recall, accuracy, the F1-score, and Matthew’s correlation coefficient (MCC).

* The article is interesting and is written with technically correct text. Clear and unambiguous, professional English is used in general, but some further improvements can be implemented to avoid redundancy, as in line 142, where it says that details about DL techniques "were not provided in the provided text", can, for example, be reduced to "DL techniques were not provided".
* The article can benefit from an additional round of writing polishing, as some typos still exist in the document, for example "CN)" in line 267.
* The article has a professional article structure and the data and Python scripts were shared and accounted in the article.
* The use of references should be improved by properly quoting the articles, as for example in line 52 it should be "(Sun et al., 2017)", instead of "Sun et al. (2017)".
* I suggest the authors to add to their references the article of Gonzales-Martinez and van Dongen (2023), due to the reasons described below in Section 2 (experimental design).

Gonzales-Martinez, R., and van Dongen, D. M. (2023). Deep Learning Algorithms for Early Detection of Breast Cancer: A Comparative Study with Traditional Machine Learning. Informatics in Medicine Unlocked
Volume 41, 2023, 101317

https://www.sciencedirect.com/science/article/pii/S2352914823001636

Experimental design

EDITED

I agree with the experimental design in terms of the architecture of the DL models applied and compared in the study with cross-validation. The experimental design properly answers the research questions and the methods are described with sufficient detail.

As a suggested improvement, I believe that in the case of cancer detection the False Negative Rate (FNR) and the False Omission Rate (FOR) are the most relevant clinical and statistical metrics to compare cancer detection algorithms. Since FNR measures the proportion of actual positive cases of breast cancer that are incorrectly classified as negative cases, it quantifies the rate of missed positives (Type II errors), and hence a high FNR implies late detection of anomalies. FOR measures the proportion of false negative errors or incorrect omissions in a decision-making process. As FOR captures failures to detect breast cancer, it is also relevant in the comparison of machine learning and deep learning algorithms, because missing the detection of a positive condition of breast cancer can have significant health consequences for cancer patients. Thus, I suggest the authors to include these metrics in the core evaluation of their proposed models, as in Gonzales-Martinez and van Dongen (2023):

Gonzales-Martinez, R., and van Dongen, D. M. (2023). Deep Learning Algorithms for Early Detection of Breast Cancer: A Comparative Study with Traditional Machine Learning. Informatics in Medicine Unlocked
Volume 41, 2023, 101317

https://www.sciencedirect.com/science/article/pii/S2352914823001636



ORIGINAL

I agree with the experimental design in terms of the architecture of the DL models applied and compared in the study with of cross-validation. The experimental design properly answers the research questions and the methods are described with sufficient detail.

As a suggested improvement, I believe that in the case of cancer detection the False Negative Rate (FNR) and the False Omission Rate are the most relevant clinical and statistical metrics to compare cancer detection algorithms. Since FNR measures the proportion of actual positive cases of breast cancer that are incorrectly classified as negative cases, it quantifies the rate of missed positives (Type II errors), and hence a high FNR implies late detection of anomalies. FOR measures the proportion of false negative errors or incorrect omissions in a decision-making process. As FOR captures failures to detect breast cancer, is also relevant in the comparison of machine learning and deep learning algorithms, because missing the detection of a positive condition of breast cancer can have significant health consequences for cancer patients. Thus, I suggest the authors to include these metrics in the core evaluation of their proposed models, as in Gonzales-Martinez and van Dongen (2023):

Gonzales-Martinez, R., and van Dongen, D. M. (2023). Deep Learning Algorithms for Early Detection of Breast Cancer: A Comparative Study with Traditional Machine Learning. Informatics in Medicine Unlocked
Volume 41, 2023, 101317

https://www.sciencedirect.com/science/article/pii/S2352914823001636

Validity of the findings

I believe the validity of the findings and the conclusions linked to the findings should be evaluated on the basis of the lowest FNR and FOR, and not only on the accuracy of deep learning algorithms, since, as the authors properly recognized, overfitting problems may arise and product perfect accuracy, despite the efforts of the authors to reduce overfitting.

Additional comments

I will strongly advise to the Editor the publication of the article, when FNR and FOR are included in the article as additional metrics to evaluate the performance of the proposed models.

Reviewer 3 ·

Basic reporting

The article shows an important matter that has been studied by the computer vision field thoroughly.
The paper should Not state phrases like "impressive accuracy", "remarkable results " and similar.

On line 83, its not clear if the authors are referring to their research objectives or previous work. Should be clarified.
On lines 159-160, just mention figure1. Its not necessary to explain on text what is clear on figure 1.
On line 313-316 the authors mention the over fitting effect that reflects that on real data makes the model not perform well.

Experimental design

Its a standard deep learning phases method to train an test with cross validation metric evaluation.
classification results of perfect score its no an evidence that this could be replicated on new data.
This should be corrected not showing perfect classification because it reflects clearly an overfitting issue.

Validity of the findings

The main problem of the findings is not validated with new data which would be a more strong way to identify if tunning parameters would have the outstanding performs that is stated on the paper.

Additional comments

The paper is well written and adressed with good computer science background. But the perfect scoring would suggest an overfitting issue. Would be helpfull to apply the proposed model to an additional dataset to increase evidence of classification stability.

---

## Round 0.2 · Minor Revisions

Kindly revise the manuscript as per the reviewer suggestions and resubmit it.

Reviewer 1 ·

Basic reporting

The authors have addressed my issue.

Experimental design

NA

Validity of the findings

NA

Additional comments

NA

·

Basic reporting

I am glad to have a second chance to review the manuscript. I am grateful that the authors take into account my comments. I suggest the paper to be accepted for publication after making the following change: in page 14, lines 428-431 it says:

"Table 4 presents an analysis of the False Negative Rate (FNR) and False Omission Rate (FOR) [38] for a set of baseline models and a proposed model when executed on two different datasets: BreakHis and 430 WBC. This analysis provides valuable insights into the models’ performance, particularly their ability to correctly identify positive cases and minimize the risk of false negatives and omissions."

I suggest to clarify that [38] refers to:
[38] Gonzales-Martinez and Daan-Max van Dongen. "Deep Learning Algorithms for Early Detection of Breast Cancer: A Comparative Study with Traditional Machine Learning" (2023), Informatics in Medicine Unlocked, Volume 41, 101317.

And I also suggest the author to compare their results with those of Gonzales-Martinez and van Dongen (2023), in order highlight that the reason for minimizing the FNR and FOR is the high cost of false negative cases of cancer detection.

Experimental design

The experimental design is appropriate.

Validity of the findings

The findings are valid due to low FOR and the low FNR, I suggest the authors to highlight and extend more about these results.

Reviewer 3 ·

Basic reporting

The paper shows an adequate context using new information
I take note of changes an find this version more accurate.

Experimental design

Two dataset are presented which i consider a benefit for readers.

Validity of the findings

Metrics presented are consistent with the methods used on similar contexts.

---

## Round 0.3 · accepted · Accept

Author has addressed reviewer comments properly. Thus I recommend publication of the manuscript.